# Charging Protocol for Partially Rechargeable Mobile Sensor Networks

**DOI:** 10.3390/s23073438

**Published:** 2023-03-24

**Authors:** Li-Ling Hung

**Affiliations:** Department of Computer Science and Information Engineering, Aletheia University, New Taipei 25103, Taiwan; llhung@au.edu.tw

**Keywords:** partial rechargeable mobile sensor network, energy harvesting, rechargeable sensor, energy transmission, network lifetime

## Abstract

Wireless sensor networks (WSNs) have wide applicability in services used in daily life. However, for such networks, limited energy is a critical issue. The efficiency of a deployed sensor network may be subject to energy supply. Wireless rechargeable sensor networks have recently been proposed and discussed. Most related studies have involved applying static rechargeable sensors to an entire rechargeable environment or having mobile chargers patrol the environment to charge sensors within it. For partially rechargeable environments, improving the recharge efficiency and extending the lifetime of WSNs are considerable challenges. Scientists have devoted attention to energy transmission technologies and mobile sensor network (MSN) applications. In this paper, we propose a flexible charging protocol in which energy can be transmitted from certain energy supply regions to other regions in an MSN. Mobile rechargeable sensors are deployed to monitor the environment. To share energy in a certain region, the sensors move to replenish their energy and transmit energy to sensors outside the energy supply region. The efficiency of the proposed protocol is also discussed in the context of various situations. The evaluation results suggest that the flexible protocol is more efficient than other charging protocols in several situations.

## 1. Introduction

Numerous wireless sensor network (WSN) applications, such as those for military use, surveillance, environmental monitoring, and health care, have been proposed in the literature [1,2,3]. Monitoring lifetime is a critical factor for WSN applications, and it is dependent on the first sensor exhausting its energy; such sensors are usually equipped with batteries with limited energy. Thus, limited sensor energy is a crucial challenge in the design of WSNs. To alleviate such energy limitations, many researchers have proposed energy-efficient routing mechanisms for WSN applications [4,5,6,7]. In addition, some researchers have proposed applying different sensor duty cycles to prolong the lifetime of WSNs [8,9]. Recently, wireless rechargeable sensor networks have been proposed [10,11,12,13].

Sensor energy harvesting has been proposed to extend such sensors’ monitoring lifetime and maximize their utility [14,15,16,17,18,19,20]. Related energy sources include electromagnetic fields, radio frequency (RF) energy, solar energy, and combined energy sources [15]. Some researchers have proposed an energy scavenging approach that involves replenishing energy from the environment [21,22,23] under the assumption that the energy supply is distributed equally throughout the entire environment. However, in the natural environment, green energy may be supplied only in particular regions. Thus, an energy harvesting mechanism that assumes that the energy supply is confined to a particular region is a practical solution.

Wireless energy transmission approaches have been proposed to overcome this energy supply problem [11,16,17]. Energy transmission improves the energy supply in a particular region, enabling such energy to be transmitted to sensors in other regions of the monitored environment. Kafle et al. [24] proposed that energy transmission is similar to data transmission over the Internet. However, differences exist between the two; during data transmission, the receiver receives the same data that the sender sends, but during energy transmission, the receiver receives less energy than what the sender sends due to propagation loss.

The Internet of Energy has been proposed in recent years [24,25,26]. Researchers have mentioned that smart energy and future energy networks are depending on telecommunications technologies. One challenge of sensor energy transmission is the integration standard.

This paper proposes mechanisms for sensors harvesting energy in specific regions of the environment and those sensors monitoring the entire environment cooperatively. With these mechanisms, rechargeable sensors can monitor the environment and cooperatively recharge themselves. Energy consumption schemes were considered under a flexible model. If the total energy supply in a partial environment is sufficient to satisfy the total energy consumption in the entire environment, the monitoring lifetime of the environment can be substantially extended.

The major contributions of this study are summarized as follows:A cooperative scheme was designed for mobile sensors to monitor the environment and harvest energy from certain charging regions.A scheme was designed for sensors to share the energy supplied in certain charging regions.A flexible two-layer recharging model was proposed. Sensors in the first layer replenish themselves by moving to charging areas, and sensors in the second layer harvest energy from neighbors in the first layer.

## 2. Related Studies

Many researchers have devoted attention to improving sensor energy efficiency; the energy of sensors in a network is limited by energy supply. To reduce related energy consumption, Agarkhed et al. [6] proposed defining different priorities for events and applying packet aggregation. An event packet is sent according to the priority defined. If the priority of an event packet is low, it may wait and aggregate with other packets. This mechanism reduces the energy consumed for data transmissions. In Hung’s [5] proposed smart routing mechanism, different transmission periods are arranged among surrounding neighbors. The use of a smart antenna could substantially improve parallel transmissions. Additionally, the mechanism increases monitoring accuracy because the monitoring areas of such sensors overlap. Kozlowski and Sosnowski [27] discussed the tradeoff between duty cycling and wake-up radio techniques for improving energy efficiency.

A sensor harvests the energy, which may then be distributed by an energy charger or scavenged from nature. To charge related sensors, researchers have proposed distributing energy efficiently by using mobile energy suppliers, otherwise termed mobile chargers [16,17,18,19,20]. Guo et al. [18], Wei et al. [19], and Ma et al. [20] have proposed that mobile chargers replenish sensors and collect monitoring data from them. However, a mobile charger may require the use of powerful and costly devices that may not be able to move in a narrow region. In addition, data collection through movement is inefficient when data are urgently required. Furthermore, Kosunalp et al. [28,29] proposed energy prediction algorithms for energy harvesting in sensor networks. Through these algorithms, these researchers have encouraged further developments related to energy scavenging from the environment.

Energy sharing in mobile networks has been proposed in recent years. Dhungana and Bulut [30] presented peer-to-peer energy sharing applications for mobile networks. Recently, researchers have proposed adding certain functions to a chip in a single device to satisfy system energy requirements [31,32]. Furthermore, Shaikh et al. [14] developed sensor nodes with in-built energy harvesting subsystems. In the present study, it is assumed that each mobile sensor equipped with a recharging battery can harvest and transmit energy.

Yi and Yoon [17] and Malebary [11] have proposed charging sensors using a mobile charger; in their approach, charging is conducted through wireless transmission. The mobile charger moves around the environment to transmit energy to the sensors it passes by and collects the data from sensors simultaneously. Lu et al. [16] introduced several mechanisms for wireless energy transmission. He et al. [33] and Fu et al. [34] have proposed that wireless recharging may reduce energy consumption from movement and recharging delay as well as increase charging efficiency. The power transmission can be derived using Equation (1) [12], where *P_t_* and *P_r_* are the transmission power and receiving power, respectively. In addition, λ and η are the carrier wavelength in meters and the rectifier efficiency of the antenna, respectively; and β and *L_p_* are parameters used to adjust the Friis free-space formula for short-distance transmissions and dimensionless antenna polarization loss, respectively. Moreover, *d* is the distance between the transmitter and receiver antennas.
(1)$$P_{r}=P_{t}\times \frac{G_{r}G_{t}\eta }{L_{p}}{\left(\frac{\lambda }{4\pi (d+\beta )}\right)}^{2},\;\mathrm{Where}\;\frac{G_{r}G_{t}\eta }{L_{p}}{\left(\frac{\lambda }{4\pi (d+\beta )}\right)}^{2}\mathrm{is}\;\mathrm{the}\;\mathrm{antenna}\;\mathrm{gain}$$

Han et al. [35] revealed that $\frac{G_{r}G_{t}\eta }{L_{p}}=4.32\times 10^{-4}$ and β=0.2316. We refer to the difference between *P_t_* and *P_r_* as the propagation loss. Yi and Yoon [17] showed that using inductive coupling, the transmission efficiency is 90% and 40% at distances of 1 and 2 m, respectively. Moreover, the efficiency is higher than 90% when the distance is in the millimeter range using magnetic resonance coupling; when using RF transmission energy, the energy efficiency is 50% at most. Furthermore, Guimaraes et al. [12] demonstrated that omnidirectional antennas or directional antennas can be used for charging. Therefore, when a suitable arrangement of appropriate antennas is used for data or energy transmission (e.g., different types of antennas for different purposes or the same type of antennas in different channels), wireless energy transmission may be more efficient and thus more widely used than it was before. In our design, the antennas for basic functions of sensors, including event detection and data transmission, are different from those for energy transmitting or receiving. Our design does not interfere with the manipulation of basic functions but takes the energy consumption for the manipulations into account. We may use another antenna system for energy transmission in which the sensor stays or use magnetic coupling in which sensors move close for transmission.

Kafle et al. [24] introduced an energy router that can harvest energy and dispatch energy to achieve energy balance. We consider mobile sensors to be energy routers that can dispatch an appropriate amount of energy, and that energy can flow between each pair of mobile sensors through energy transmission. Wireless energy transmission has two forms. In the first form, energy is transmitted to a single device; in the second form, energy is transmitted to multiple devices simultaneously [36]. In addition, the charging efficiency of a single device when multiple devices are being charged is lower than that when one device is being charged. However, when multiple devices are being charged, the total charging efficiency in the environment is better than that when a single device is being charged and increases linearly with the number of devices. According to the law of energy conservation, the total amount of energy received must not be greater than the amount of energy transmitted.

According to the antenna gain shown in Equation (1), a long distance between the energy transmitter and the energy receiver reduces transmission efficiency. In addition, wireless energy transmission from a certain energy supply to other regions in the environment is feasible when the distance is short. The energy transmission distance may increase when transmission technology breakthroughs are achieved. Through movement, mobile sensors can shorten the related transmission distance.

Mobile sensors expend some energy in moving to recharge themselves or neighboring sensors. Such energy expenditure will be reduced with the development of new material techniques; lighter devices expend less energy than heavier ones do. During such transmissions, sensors are also hampered by propagation loss, and transmission technology developments might alleviate this problem. In consideration of the rapid development of related technologies, this study focused on the development of mobile sensor cooperation models for a partially rechargeable environment. A flexible mechanism can be applied to balance energy consumption from sensor movement and energy transmission depending on material and energy transmission technologies. This study focused on the feasibility of applying partial rechargeable sensor networks in the future.

## 3. Architecture of Charging Protocols

### 3.1. Network Environment and Problem Formulations

The monitored area was divided into numerous equal-sized hexagons, termed cells in this paper. The edges of each cell have a length of *r*, which can be set as one-half of the sensing distance of a sensor. Assume that in a monitoring environment, *k* energy supply zones (also called charging zones) exist for charging sensors (*Z* = {*z_l_*| *l* = 1, …, *k*}). Rechargeable sensors can harvest energy in the charging zones, and the environment can be divided into *k* recharging zones. An energy supply is a mechanism that can collect energy from an energy source, for example, fire, sun, wind, or others, and can charge sensors using collected energy. The energy supplies are deployed in the charging zone. For example, in an intense sunlight zone, a sensor equipped with a solar panel may harvest energy in the zone wirelessly and the intense sunlight zone is the charging zone or named charging region. In order to extend the lifetime of the sensor network, the sensors harvest energy from the charging region and then transmit the energy to sensors in other regions. 

The division of recharging zones can be based on the amount of energy supplied in such zones. Figure 1 shows an environment with three energy supply (or charging) zones. The environment is divided into three recharging zones, named zones A, B, and C, marked by red lines. For event detection and wireless energy transmission, *p* rechargeable mobile sensors are available (*S* = {*s_j_*| *j* = 1, …, *p*}). In the system, the residual energy, the location of each sensor, and the location of the charging zone in each sensor’s area are known. For simplification of presentation, after the recharging zones for each supply zone were divided, an energy transmission scheme for each divided recharging zone was constructed.

### 3.2. Coordinate Origin Orientation

Each recharging zone is ascribed to a coordinate system and can be divided into several hexagonal cells. Although the sensors are deployed in a plane environment, according to the characteristics of hexagonal cells and for ease of locations’ definition, we defined three coordinate axes that divide each recharging zone into six regions. For example, regions A, B, and C in Figure 1 have their own coordinate systems. The coordinates (0, 0, 0) are established by calculating the ratio of the amount of energy supplied in the charging zone to the amount of energy required in the recharging zone. The horizontal line (i.e., the Z-axis) crosses the origin (0, 0, 0); we turn the Z-axis 60° clockwise such that it becomes the X-axis, and the X-axis is turned 60° clockwise to become the Y-axis. The location of each cell is represented by a coordinate (*x*, *y*, *z*). The points to the right of the origin have positive values with regard to the axes, and the distance unit is 3*r*/2. According to the definitions, the points on the X-, Y-, and Z-axes have the same characteristics as y+z=0, x−z=0, and x+y=0, respectively, as shown in Figure 2 and Figure 3. The area is divided into six areas by these axes (A_1_ through A_6_), and they include energy supply zones Z_1_ through Z_6_. The farthest cells away from the origin in Z_1_ through Z_6_ named cells *c*_1_ through *c*_6_. The cells in A_1_, A_2_, A_3,_, A_4_, A_5_, and A_6_ satisfy x−z>0 and y+z>0, x−z<0 and x+y>0, x+y<0 and y+z>0, x−z<0 and y+z<0, x−z>0 and x+y<0, and y+z<0 and x+y>0, respectively. The operation model for sensors in these six zones have similar designs, but the rules of their operation may be altered due to differences between coordinates.

For appropriate origin orientation, we adjusted the axes according to the energy required in the recharging regions. The objective of this adjustment was to balance energy requirements and energy supply. Let |R| represent the size of area R; the size of the energy supply area should be proportional to that of the area in which energy is required. This stipulation is represented as follows: |Z_1_|:|Z_2_|:|Z_3_|:|Z_4_|:|Z_5_|:|Z_6_| ≈ |A_1_|:|A_2_|:|A_3_|:|A_4_|:|A_5_|:|A_6_|.

First, the origin was set as the center of the charging zone temporarily, and the amount of energy required in each area was calculated. Subsequently, in order to balance the differences among the ratios of energy requirements to energy supply, we shifted the axis with the largest gap of energy requirements on both sides of the axis to reduce this gap and then updated the coordinates of cells and the information on the amount of energy required in the regions. For ease of calculation, cells c1 through c6 are employed to derive the shifted distance. After the first adjustment, a second adjustment similar to the first one was conducted; the axis with the largest gap was to be shifted. Subsequently, the last axis is shifted to cross the intersection of the previous two axes, and the intersection is the new origin. Figure 2 provides an example of such adjustments. Figure 2a shows the initial state in which the origin is at the center of the charging zone surrounded by red lines which are separated into six same-size regions. However, three axes split the recharging area into ratios of 8:10, 7:11, and 5:7, respectively. The largest gap, 7 and 11, is formed by Y-axis, so the Y-axis was selected first for adjustment. The Y-axis was shifted left to make the ratio of distance $\bar{c_{6}c_{d}}$ to distance $\bar{c_{d}c_{3}}$ similar to 7:11 (see Figure 2b). Figure 2b shows that the line originally represented x − z = 2 will be the new Y-axis. After the Y-axis was adjusted, we had the equation: |Z_1_ + Z_2_ + Z_3_|:|Z_4_ + Z_5_ + Z_6_| ≈ |A_1_ + A_2_ + A_3_|:|A_4_ + A_5_ + A_6_|.

Subsequently, the Z-axis was selected for the second adjustment (see Figure 2c). While adjusting the Z-axis, the line originally represented x + y = 1 will be the new Z-axis and we had the following equation:|Z_2_ + Z_3_ + Z_4_|:|Z_5_ + Z_6_ + Z_1_| ≈ |A_2_ + A_3_ + A_4_|:|A_5_ + A_6_ + A_1_|. Subsequently, the X-axis was adjusted to fix the origin at the intersection of the Y- and Z-axes [Figure 2d]. After adjustment, the energy harvested in area Z*i* supported the energy consumed in area A*i*. Usually, a large energy requirement in A*i* matches a large charging area in Z*i*. The coordinates of cells located in the charging zone presented in Figure 2d are shown in Figure 3.

To balance the energy in the environment, the energy transmission of each sensor is calculated. To improve the energy transmission efficiency, the energy transmitted and received among sensors can be arranged in advance. In this paper, we propose an energy transmission model for partially rechargeable MSNs. In the model, the charging zones are areas containing energy suppliers. Each energy supply region forms a coordinate system, including charging zones and recharging zones, and provides energy to sensors in the coordinate system. Each sensor knows its coordinates, the location of its energy supply zone, and the borders of the recharging zone.

### 3.3. Sensor Energy Model

This aim of this study was to employ the energy available in certain regions to extend the lifetime of a WSN. Under the assumption that the number of sensors used is sufficient for monitoring the entire environment, the lifetime of a WSN depends on the energy of the sensor with the shortest lifetime. This study focused on sensor energy replenishment or energy transmission under the condition of full-coverage monitoring. A sensor may move to replenish its energy, transmit energy to other sensors, or monitor the environment. We assume that in charging zones, the amount of energy supplied is greater than what the sensors consume for detection and movement in that zone; moreover, they may gain energy, enabling them to transmit energy to a recharging zone.

To prolong the lifetime of a WSN, the energy in the supply zone should be transmitted to recharging zones. Let $E_{i}^{t}$ be the residual energy of sensor *s_i_* at the *t*th time slot, where $H_{i}^{t}$ and $C_{i}^{t}$ are the harvested and consumed energy of *s_i_* during the *t*th time slot, respectively. The energy of *s_i_* at the *t*th time slot is shown in Equation (2). The value of $H_{i}^{t}$ can be derived using Equation (3), where $HS_{i}^{t}$ and $HF_{i}^{t}$ represent the energy harvested from suppliers and from the neighbors of *s_i_* in the *t*th time slot, respectively. The value of $C_{i}^{t}$ can be derived using Equation (4), where $CM_{i}^{t}$ and $CF_{i}^{t}$ represent the energy consumed by *s_i_* in the *t*th time slot for movement and energy transmission, respectively. In addition, $CT_{i}^{t}$, $CD_{i}^{t}$, and $CS_{i}^{t}$ are the energy consumed by sensor *s_i_* in the *t*th time slot for message transmission, event detection, and standby, respectively. In the proposed protocol, a sensor may consume energy for movement or transmission. Each sensor must consume energy for event detection and message transmissions. When *s_i_* is located in a charging zone, $HS_{i}^{t}=\alpha_{i}^{t}\times H$, where *H* is the amount of energy harvested by a sensor during a certain time slot in the charging zone, and it is assumed to have a constant value. When *s_i_* is not located in the charging zone, the values of $\alpha_{i}^{t}$ and $HS_{i}^{t}$ are 0 because they cannot obtain energy from the energy supplier in the charging zone.
(2)$$E_{i}^{t}=E_{i}^{t-1}+H_{i}^{t}-C_{i}^{t},\;E^{full}\geq E_{i}^{t}\geq 0\;\mathrm{and}\;\forall s_{i}\in S$$
(3)$$H_{i}^{t}=HS_{i}^{t}+HF_{i}^{t}$$
(4)$$C_{i}^{t}=CM_{i}^{t}+CF_{i}^{t}+CT_{i}^{t}+CD_{i}^{t}+CS_{i}^{t}$$

Generally, due to propagation loss, the amount of energy obtained is smaller than that transmitted. Therefore, when *s_i_* transmits energy to *s_j_*, the value of $CF_{i}^{t}$ is larger than that of $HF_{j}^{t}$. The propagation loss for wireless transmission, *PL_w_*, is derived using Equation (5).
(5)$$PL_{w}=CF_{i}^{t}-HF_{j}^{t}$$

We assumed that the number of sensors was adequate for the extent of monitoring required. When a sensor reaches an energy value of 0, the lifespans of that sensor and its sensor network end. The main objective of this study was to maximize the lifetime of sensor networks by optimizing the use of energy from a particular region of an environment, as shown in Equation (6).

Objective:(6)$$\mathrm{Maximize}\;T=l\times d,\;\mathrm{subject}\;\mathrm{to}\;E_{i}^{p}>0,\;\mathrm{for}\;\forall s_{i}\in S\;\mathrm{and}\;p\leq l$$
where *d* and *l* are the time duration of each round and the number of rounds, respectively. Let |*T*| denote the maximal round of *T* or maximal integer *T*/*d*.

Let ${\bar{EH}}_{0}$ and ${\bar{EC}}_{0}$ be the total amount of energy harvested by sensors in the charging zone and consumed for environment monitoring, respectively, including event detections and data transmissions, during time *T*. As stated, energy supply is only available in the charging zone. The harvested energy of sensors in the environment is shown in Equation (7), where $\alpha_{i}^{t}$ is 1 when *s_i_* is in the charging zone; otherwise, the value of $\alpha_{i}^{t}$ is 0. As mentioned in the previous section, *H* is the amount of energy harvested by a sensor in the charging zone during a certain time slot. Thus, ${\bar{EH}}_{0}$ can be derived as n×H×|T|, where *n* is the number of sensors in the charging zone, if the number of sensors in the charging zone is constant. ${\bar{EC}}_{0}$ represents the energy consumption of sensors, such as for event detection, data transmission, and idle waiting, as shown in Equation (8), where $\beta_{i}^{t}$ is 1 when *s_i_* transmits a message in the *t*th time slot; otherwise, $\beta_{i}^{t}$ is 0. When the total energy supplied exceeds the total energy consumed by the sensors, the harvested energy can be used to extend the lifetime of a sensor network.
(7)$$\bar{EH_{0}}=\sum_{t=1}^{\left|T\right|}\sum_{i=1}^{\left|S\right|}\alpha_{i}^{t}H,\;\mathrm{where}\;s_{i}\in S$$
(8)$$\bar{EC_{0}}=\sum_{t=1}^{\left|T\right|}\sum_{i=1}^{\left|S\right|}(\beta_{i}^{t}CT_{i}^{t}+CD_{i}^{t}+CS_{i}^{t}),\;\mathrm{where}\;s_{i}\in S$$

When the sensors near the charging zone transmit energy to sensors far away from this zone, the consumed energy includes the movements of sensors and the energy lost for propagation. Therefore, the energy consumed is shown in Equation (9), where $\alpha_{i,j}$ equals 1 when *s_i_* transmits energy to *s_j_* in the *t*th time slot, and $\gamma_{i}$ equals 1 when *s_i_* is moving in the *t*th time slot. $CF_{i}^{t}$ is the amount of energy that *s_i_* transmits in *t*th time slot, and $HF_{j}^{t}$ is the amount of energy that *s_j_* received in the *t*th time slot. Thus, when *s_i_* transmits energy to *s_j_*, ($CF_{i}^{t}-HF_{j}^{t}$) is the propagation loss for energy transmission from *s_i_* to *s_j_*. $\bar{EC}$ in Equation (9) summarizes the energy lost for energy transmissions and the energy consumed for movements in the entire environment. If the total amount of energy harvested by sensors (${\bar{EH}}_{0}$) exceeds the energy consumed for environment monitoring ($\bar{EC}$), then the recharging efficiency is positive, and the lifetime of the sensor network can be extended. The amount of energy in the network at time *T*, *E*(*T*), is obtained using Equation (10), where *E_init_* is the initial energy in the network. *E*(*T*) represents the residual energy in the environment at *T*.
(9)$$\bar{EC}=\sum_{t=1}^{\left|T\right|}\sum_{j=1}^{\left|S\right|}\sum_{i=1}^{\left|S\right|}(\alpha_{i,j}\times (CF_{i}^{t}-HF_{j}^{t}))+\;\sum_{t=1}^{\left|T\right|}\sum_{i=1}^{\left|S\right|}\gamma_{i}CM_{i}^{t},\;\mathrm{where}\;s_{i},s_{j}\in S$$
(10)$$E(T)=E_{init}+\bar{EH_{0}}-\bar{EC}-\bar{EC_{0}}$$

Because sensor energy is limited and the energy supplier is located within a particular zone, the sensors cooperate to transmit energy to other regions. To transmit energy over the entire environment, sensors inside the charging zone transmit energy to those outside the zone. One challenge of the proposed distribution mechanism is energy-efficient cooperation between sensors. In this study, we proposed a flexible model that can be adjusted to deal with different situations. Moreover, the energy harvested and consumed in such situations and the rules for extending the lifetime of a sensor network is inferred. As we mentioned in related studies, a shorter energy transmission distance reduces propagation loss; in a few millimeters, the energy transmission efficiency can be more than 90% with inductive coupling or with magnetic resonance coupling. In order to employ energy efficiently, in the proposed model, the sensors transmit energy using near-field radio frequency transmission technology or magnetic coupling. That is, sensors need to move to transmit energy at a close distance.

## 4. Proposed Charging Protocol

It was assumed that the sensors in charging zones harvest energy from energy suppliers and that sensors in recharging zones harvest energy from neighbors. Furthermore, it was assumed that sensors harvest energy more than they consume, either from movement or energy transmission when they are in charging zones. Moreover, when a sensor transmits energy to its neighbors, it must retain energy for its continued operation, and propagation loss results in the amount of energy obtained is less than the transmitted energy. As Equations (2) and (4) mentioned, the operations of a sensor include event detection, data transmission, and standby; these are considered in its energy consumption. Because the energy consumed for the basic functions of sensors is not affected by charging protocols, the energy consumed for the basic functions is considered a constant which is calculated by time, for example, 10 mJ/s or 20 mJ/s. In this study, we considered building a general model which transmits energy from the charging zone to the recharging zone.

### 4.1. Overview

We propose a flexible charging protocol, named FCP, for harnessing the energy supplied in a particular region to extend the lifetime of sensor networks. The FCP can be applied after the origin of the coordinates is orientated. The protocol divides the environment into movement and energy transmission regions. Sensors move in the movement region following defined paths and replenish their energy in the charging zone. When arriving at the border of the movement region, sensors transmit energy to the sensors on the border. Moreover, in the energy transmission region, sensors move in their own hexagonal cells to harvest energy from neighbors and spread energy to their forwarding neighbors. Table 1 presents the pseudocode for the proposed protocol.

As detailed in the pseudocode, the first step of the FCP is to determine the coordinates of the systems in the movement and energy transmission regions. The second step is to determine the movement paths of sensors in the movement region or the transmission paths of those in the transmission region. The roles of sensors depend on their locations. After sensors obtain information on their roles, they execute related activities in the third phase. Details of the second and third phases are based on the locations and roles of sensors. Detailed descriptions according to the roles of sensors are presented in Section 4.2 and Section 4.3. The first phase is described in the subsequent paragraphs.

The range of the movement region can be selected to be *f* times that of the charging zone, extending from the center of that zone. The value of *f* can be determined by the energy capacity of the sensors. An appropriate *f* value can be selected when the protocol is applied in practice. For ease of presentation, in the following descriptions, we set the value of *f* to 2 (i.e., the distance between the origin and the border of the movement region is twice that between the origin and the border of the charging zone). The sensors in the other region harvest the energy transmitted from the movement region.

The sensors in the movement region cooperate with those monitoring the environment as they move. Each cell is always monitored by one sensor because when one sensor (a previous sensor) leaves the center of a cell to move to an adjacent cell, another sensor (the subsequent sensor) moves toward the center of the cell the previous sensor is departing and can monitor its environment. Moreover, when sensors in the energy transmission region move to transmit and harvest energy, the cells should be monitored by sensors before and after sensor movement. Therefore, the divided cells in the energy transmission region are smaller than those in the movement region. Assuming the edges of the hexagonal cell in the movement region are denoted by length *R*, as previously mentioned, the length *r* is one-half of the sensing distance; for the monitoring environment, the value of *R* cannot be greater than 2*r*.

Figure 4 provides an example of different coordinates in the environment. In Figure 4, the area within the blue circle is the charging zone, and that within the green circle is the movement region. The red cells are in the movement region, and the black cells are in the energy transmission region; the black cells that overlap with the movement region are ignored when discussing cells in the energy transmission region. Both red cells and black cells have their own coordinate systems. The coordinate systems of the movement and transmission regions can be formed from the same origin, even though the cell sizes are different. The coordinates of the movement region are used to arrange the movement paths, and those of the transmission region are used to determine the previous cells and subsequent cells. The yellow region in Figure 4b is the intersection of the movement and transmission regions. The sensors move in the movement region and transmit energy to the sensors in the intersection.

The sensors in the movement region replenish themselves in the charging zone and move to the intersection. When located in the intersection, they transmit energy to neighboring sensors with less energy than themselves. The sensors in the transmission region move in their cell to harvest energy from their upstream sensors or transmit energy to their downstream sensors. Note that, when moving or transmitting energy, the sensors detect events and transmit data simultaneously because the antenna systems for basic functions are independent of movements and energy transmissions.

### 4.2. Sensors in the Movement Region

Although sensors are deployed in the entire environment because their energy is limited and the energy supply is located only in certain regions, sensors cooperatively monitor the environment and replenish their energy. The sensors in the charging zone move around to replenish themselves and transmit energy from the movement region to sensors in other regions. To improve cooperation in sensor monitoring and reduce energy consumption related to cooperation communications, the movement paths and transmission rules of sensors are defined in advance. On the basis of its location, each sensor determines its next step in accordance with defined paths and time slots.

A set of movement paths are defined for sensors moving from a charging zone to the borders of a movement region and back to the charging zone. For ease of presentation, we name the subsequent cell and previous cell in a path as the next and last cell, respectively. These movement paths start from the origin and move along the centerlines of regions (e.g., line *x* = 0 in A_2_ of the proposed coordinate system, as shown in Figure 5). To achieve an energy balance in the environment, when sensors arrive at the border of a movement region, each path branches into two paths to allow sensors to move back to the charging zone. One path passes through the region on the left of the centerline, and the other passes through the region on the right of that line. With a branch point as a pivot, each sensor knows its path according to the time slot in which it reaches the pivot. When a sensor reaches the *k*th time slot and *k* is an odd number, it moves through the path on the left. Otherwise, it moves through the path on the right. We assume that a time slot includes the time period in which a sensor stays in a cell and moves from that cell to an adjacent one. With coordinates and rules for movements, sensors can identify their paths according to their locations and time slots. For example, Figure 5 shows the movement paths in A_2_. Because the movement path branches into two paths after sensors move along the centerlines to the pivot, the frequencies of movements by sensors in other locations, except the centerlines, are half of those in the centerlines (i.e., the sensors remain in cells for a longer time before moving to the next cell).

A sensor’s energy increases when it moves into a charging zone. In addition, sensors transmit energy when they are on the border of movement regions. Because sensors must monitor the environment and continue moving along a path, they reserve sufficient energy for monitoring and returning to charging zones. Sensors possess their lowest energy when they are entering a charging zone. When they enter this zone, their energy increases. Regardless of whether a sensor is stationary or moving, it harvests energy when it is inside the charging zone. However, when sensors transmit energy to neighbors, these sensors should not be far away (i.e., they should be in the millimeter range or almost touching).

### 4.3. Sensors in the Energy Transmission Region

To extend the lifetime of sensor networks, the sensors around charging zones transmit energy to those far away from such zones. The sensors in the transmission region cooperate by transmitting energy wirelessly over a short distance. Herein, a transmitting sensor is called an upstream sensor and a receiving one is called a downstream sensor. To achieve energy transmission efficiently, sensors identified upstream and downstream sensors and the amount of energy required in advance. Different from moving around the movement region, the sensors in the transmission region move over a much shorter distance, which is less than 2*r*. As they move in order to transmit energy to sensors at a close distance, the short distance is good for energy transmission.

The energy transmission routes can be determined by finding the shortest paths from the movement region’s border to the transmission region’s border. The downstream and upstream sensors are identified when the energy transmission paths are formed. In formed transmission paths, each sensor may have one or more downstream sensors but only one upstream sensor.

Moreover, sensors should know the amount of energy required for energy transmission to downstream sensors. After transmission routes are formed, at the end of each route, a sensor sends a notification of a count *c* with a value of 1 (i.e., *c* = 1) to its upstream sensor. When a sensor receives a count value from its downstream sensor, it sends a notification containing the value after it increases above 1 (i.e., *c* = *c* + 1) to its upstream sensor. If the sensor has two downstream sensors, it sends a notification containing said value after it is summarized and increased. The notification of each route stops when the notified sensors are on the border of a movement region; the counted values denote the energy requirement of the downstream sensors. Figure 6 shows the transmission routes and accounts in A_2_. The region in green is the movement region, and the darker shade represents the border. The red lines represent the transmission routes, and the values in cells represent the accounts, as the energy requirements.

After sensors in the transmission region have sufficient energy to satisfy their requirements and those of downstream sensors, they alternate between transmitting and harvesting energy. When a sensor meets its upstream sensor, it harvests energy; when it meets its downstream sensor(s), it transmits energy. To transmit and harvest energy efficiently, the transmitting and harvesting times of sensors are scheduled by their coordinates. For a sensor located at coordinate (*x*, *y*, *z*) with a *k* value equal to (|*x|* + |*y|* + |*z|*)/2, when *k* is odd, the sensor transmits energy in odd time slots and harvests energy in even slots. By contrast, when *k* is even, the sensor transmits energy in even time slots and receives it in odd slots. When transmitting energy to and harvesting energy from neighbors, the sensors move toward corresponding neighbors; they then meet and transmit energy at the edges of cells. Figure 7 shows examples of the locations of sensors while transmitting energy. Figure 7a shows the transmissions in odd time slots, and Figure 7b presents the transmissions in even time slots. The dotted lines indicate the movement trails of these sensors. The sensors on the transmission border need not move because they do not have downstream sensors to which they could spread energy. Node A in cell C of Figure 7a,b can be compared. The locations of sensors in the transmission region are on the edges or corners of cells but not in the center of them. Because the length of the hexagonal cells in the region is half the length of the sensing range, the entire cells are monitored at any time.

Sensors receive energy from upstream sensors and transmit energy to downstream sensors according to the appropriate time slots. Therefore, energy is transmitted from the movement region to the indicated recharging zones. When the amount of energy supplied is sufficiently large to cover that expended during transmission, the lifetime of the sensor network is extended. As noted in related articles [32], energy can be transmitted to one or more receivers, and when a sensor has more downstream sensors (at most two in our design), it transmits such energy to these downstream sensors simultaneously. 

During energy transmission, the amount of energy received by a sensor depends on the duration of energy transmission. Different from the data transmission, the time slot of energy transmission and movement is measured in hours. The shorter time for sensors transmitting energy and moving represents the sensors spending more energy for movement and transmission but not their basic function. For practical applications, the time slots for movement arrangement could be longer when the energy storage of sensors is larger. Also, it could be shorter when the technique of energy replenishment and energy transmission is advanced. Furthermore, because sensors’ movements in movement region are arranged at the same time, each cell in the region must be monitored by a sensor; when a sensor leaves out to its next cell, there must exist another sensor moving from its previous cell. On the other hand, because the sensors’ movements in the transmission region are arranged in their own cell only, as previously mentioned, the length of a transmission cell edge, *r*, is one-half of the sensing distance; the entire transmission cell is monitored by the sensor located in any position of this cell. All the manipulations for energy harvesting and transmission do not interfere the basic functions of sensors.

## 5. Evaluations and Discussion

This section presents evaluations and a discussion of scenarios under which the FCP is used. The harvested energy and consumed energy are derived in simulations, which are detailed subsequently.

According to the kinetic energy formula, the energy required to move a sensor is $mv^{2}/2$, where *m* and *v* are the weight and velocity of the sensor, respectively. In addition, the velocity of a sensor can be obtained using its moving distance, *d*, and moving duration, *t* (i.e., d=v×t). Thus, the energy consumption of *s_i_* from moving can be written as Equation (11). The consumption is determined by the weight of a sensor and its moving distance in a certain time period.
(11)$$CM_{i}^{t}=\frac{1}{2}m_{i}{\left(\frac{d}{t}\right)}^{2}$$

Assuming that the edges of hexagonal cells in the transmission and movement regions are *r* and *R*, respectively, the distance of each movement by sensors in the movement region is $\sqrt{3}R$, and the longest distance of each movement by sensors in the transmission region is $\sqrt{3}r$. The longest distance of each movement by sensors at the intersection is $\sqrt{3}r/2$. When sensors move toward their target locations, the velocity ratio for sensors in the movement region and transmission region is *R*:*r*. Therefore, the ratio of movement-related energy consumed by sensors in the movement and transmission region is *R*^2^:*r*^2^. For example, in the coordinate systems in Figure 4, the edges of cells in the movement and transmission region are 20 m and 10 m, respectively. Assuming that the weight of a sensor is 200 g and the movement duration for each step is 20 min, the energy consumed by the sensor movement and transmission regions is 33.36 mJ and 4.2 mJ, respectively, over 20 min. Thus, over 20 min, the energy consumed by sensor movement in the movement region is 638 mJ, and that for sensor movements in the transmission region is 180 mJ. Therefore, if the energy harvested in the charging zone is greater than 638 mJ over 20 min, such harvested energy can be transmitted, and the lifetime of the Rechargeable Mobile Sensor Network (RMSN) can be extended. When the energy harvested in the charging zone is greater than 818 mJ over 20 min, the lifetime of this RMSN can be prolonged for a longer period.

Transmission efficiency can be higher than 90% in a short distance when using inductive coupling or magnetic resonance coupling. In protocol FCP, the distance between two transmitting sensors is a few millimeters; we simulate the efficiency at 90%, 94%, and 98%, respectively.

We evaluated the energy and lifetime of sensors with different weights and with different ratios of movement region to transmission region. We measure the energy consumed with different energy transmission efficiency according to the transmission distance. The parameters in the simulation are listed in Table 2. Sensors have information on the movement distance and energy transmission direction. In the simulations, for ease of presentation, we name the result of the protocol with the name of the independent variable and related value as Name(-VariableValue)*; for example, FCP-E1 refers to protocol FCP with available energy 1J per second. Unless otherwise specified, the energy transmission efficiency is 94% and the sensor weight is 200 g; the ratio of the movement region to the entire environment is 0.5, the value of *R*/*r* is 1.7, and the available energy in the charging zone is 1.5 J/s. Therefore, the FCP is the same as FCP-E1.5-M0.5-R1.7-W1-T94.

We first evaluated the proposed protocol by comparing it with three previous mechanisms termed CRM [37], EBM [38], and MCP [17]. All of these protocols are spreading energy to the sensors in the monitored environment. The CRM is a cooperative mechanism; when a mobile sensor needs to replenish its energy, it requests that one of its neighbors helps monitor the environment. In EBM, mobile sensors move around the entire environment following defined paths. They replenish their energy when they move into a charging zone. In MCP, a mobile charger is moving around to replenish some anchor sensors around the moving path. Then the anchor sensors transmit energy to other sensors. If there exists a gathered charging region, not forming a circular path, to replenish the mobile charger itself, the moving path must be much longer than its original design and the duration for spreading energy to other sensors must also be longer. Figure 8 shows the simulation results when the support energy is 1.5 J in the charging zone. The red dotted line represents the lifetime of monitoring ended because of some sensors’ energy exhaustion. The sensors that use EBM consume more energy than others because this mechanism involves the most sensor movement. The sensors using CRM must communicate with their neighbors to ensure cooperation; such sensors expend some energy on communication. Because the number of moving sensors is lower than in the other two mechanisms, the sensors in CRM have more residual energy than those in other mechanisms. However, energy consumption on communication is much greater when more sensors need to move to the charging zone. Moreover, some sensors exhaust their energy resources because they do not have sufficient time to communicate with neighbors and move to a charging zone after 1000 min. The results of energy evaluations show that our mechanism outperformed the aforementioned mechanisms. Therefore, under the condition of gathered charging zone, one mobile charger to replenish the other sensors is not the optimal choice. Protocols CRM, EBM, and FCP have the same condition that spread the energy of a gathered changing zone out. To normalize these protocols, FCP without movement region is similar to CRM, and FCP without transmission region is similar to EBM. Thus, protocol FCP integrates protocols CRM and EBM and has greater performance than those two.

Considering the energy balance of the environment, using FCP, the sensors near the charging zone have more energy than those in other regions. The energy situation of sensors shows the energy spreading from the energy charging zone and is balanced. Using MCP, the sensors far away from the mobile charger have less amount of energy, and the sensors near the mobile charger have a much amount of energy. Figure 9a,b show the snapshots of energy amount in the environment for FCP and MCP, respectively. The energy amount for sensors using FCP is balance and the energy amount of each sensor is more than 30 mJ. On the other hand, the energy amount for most sensors using MCP is less and unbalanced. The sensors near mobile charger have more energy than those located other regions. Many sensors away from mobile charger have the energy less than 20 mJ, which is much less than those using FCP.

Figure 10a shows the simulation results when the available energy in the charging zone is 1 J/s and 1.5 J/s. We used to support the energy of 1.5 J/s in simulations FCP-M0.7, FCP, and FCP-M0.3. The ratios of the movement region to the entire environment in FCP-M0.7, FCP, and FCP-M0.3 were 0.7, 0.5, and 0.3, respectively. FCP-E1-M0.7, FCP-E1, and FCP-E1-M0.3 are similar to FCP-M0.7, FCP, and FCP-M0.3 but only 1 J/s is supported in the charging zone. When the total energy is less than 5 J in the environment, sensors with exhausted energy resources are present that cannot monitor the environment with full coverage. In other words, the lifetime of that sensor ends, and the network’s lifetime ends also. When comparing FCP1, FCP, and FCP-M0.3 with FCP-E1-M0.7, FCP-E1, and FCP-E1-M0.3, the results show that when more energy is supplied in the charging zone, the lifetime of sensor networks is extended. In addition, compare FCP with FCP-M0.3; when the ratio of the movement region to the environment is increased, the lifetime of the network is extended. However, comparing FCP and FCP-M0.7; because the energy consumed for movement is much greater than that sacrificed due to propagation loss, when the ratio of the movement region to the environment is increased, the lifetime is not increased with a higher ratio of charging zone. The lifetime of FCP-E1-M0.3 ends after 1500 min. We evaluated variations in the energy consumed and deployment costs, and the results are shown in Figure 10b. For ease of comparison, the energy consumption value is normalized to range between 0 and 500. In our evaluations, the total amount of energy supplied in the charging zone is the factor under control. However, the number of sensors may be different because of the size of cells in the environment. When using fewer sensors, the energy capacity of sensors should be larger for storing energy and transmitting energy, thus the weight and movement energy for the sensors is larger. Figure 10b shows that a high ratio of the movement region to the environment increases the energy consumption for movement but reduces the number of sensors required in the environment. The ratio of the movement region to the environment influences the number of sensors and deployment costs.

We evaluated the energy consumption due to sensor weight and variations in cell size in the movement region, and the results are shown in Figure 11a. We assume that the length of cell edges (*R*) for charging zone in simulations FCP-R2.0, FCP, and FCP-R1.5 were 2*r*, 1.7*r*, and 1.5*r*, respectively. Simulations FCP-W0.7 and FCP-W1.3 consider the different weight of sensors when the length *R* equals 1.7*r*. The sensors for FCP-W1.3 and FCP-W0.7 are 1.3 and 0.7 times the weight of 200 g, respectively. The larger cells in the charging zone, using less sensors, increase the energy consumption ratio for a sensor movement to transmission. However, because the number of sensors is reduced, the total energy consumption does not markedly increase. Figure 10a reveals that energy consumption for different sizes of cells in charging zone is not obvious. Moreover, sensor weight substantially effects energy consumption. Therefore, the weight of sensors is the key factor behind energy consumption in this model. We evaluated the performance of our protocol in terms of energy transmission efficiency (Figure 11b). In Figure 10b, FCP-T90, FCP, and FCP-T98, which used protocol FCP, had energy transmission efficiencies of 90%, 94%, and 98%, respectively. The shorter distance between transmitter and receiver has higher transmission efficiency but consumes more energy on movement. The results demonstrate that in the FCPs with lower efficiency, the amount of residual energy is smaller than that in other conditions even though they consume less energy during movement. Therefore, the impact of the energy transmission loss is greater than the energy consumed for movement. Furthermore, the weight of sensors influenced the protocol performance more obviously than did transmission efficiency. For example, the light sensors or the lighter energy transmission unit on sensors will reduce the energy consumption for movement, which will improve the energy efficiency of the protocol. Therefore, according to the results of Figure 11a,b, the performance of the proposed protocol should be improved to enhance material technologies. With advancements in technology related to material and production engineering, the performance of the protocol can be improved also.

According to the evaluations, our protocol is more energy efficient than others considering the energy propagation loss and the energy spreading speed. Comparing the performance impacted by factors, for ease of presentation, we name IA as the impact of factor A; for example, IE represents the impact of energy. The levels of performance impacted are listed as
I_E_ > I_M_ > I_W_ > I_T_ > I_R_.

The amount of energy supported and the arrangement of the movement region are two important factors to employ our protocol. The weight of sensors and the transmission efficiency impact the performance of our protocol because of energy loss and energy consumption. In addition, considering the deployment cost and the amount of available energy, we can choose an appropriate set to employ the protocol well. Furthermore, the performance of our protocol will be improved with the technological development of material and energy transmission.

## 6. Conclusions and Future Work

This paper proposes a flexible charging mechanism for charging zones in rechargeable mobile sensor networks to replenish sensor energy in the entire environment; this is achieved when sensors from charging zones share energy with sensors outside this zone. The size of the movement region and the number of sensors in the movement region can be adjusted according to the practical environment and the conditions of resources. All sensors monitor the environment and share energy. In the proposed protocol, we consider the sharing and balancing of energy in the environment when the energy supply is greater than the additional energy consumed through movement and propagation loss; in such scenarios, the lifetime of a sensor network can be extended. In this research, a flexible mechanism was applied in different situations to improve sensor charging efficiency. Evaluations related to different situations demonstrated that according to materials science and manufacturing development, the mechanism can be widely applied in real-world settings by reducing propagation loss or energy expenditure on movement. When there are charging zones that can support energy for sensors, the proposed protocol has better performance than others. When deploying a WSN using this protocol, the larger number of sensors improves extending the lifetime of sensors and the sensor network; when arranging the moving paths, the weight of sensors and the amount of energy supported should be considered. In the future, we are trying to derive general formulas for the performance according to the resources and promoting mechanisms for matching cases using an algorithm to extend the lifetime of partially rechargeable sensor networks.

## Figures and Tables

**Figure 1 sensors-23-03438-f001:**
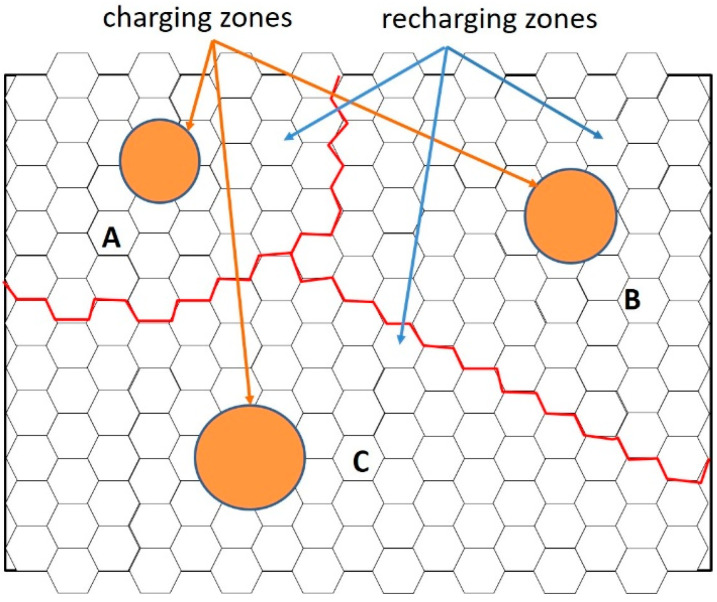
The environment is divided into zone A, zone B, and zone C according to the amount of independent energy supply zones.

**Figure 2 sensors-23-03438-f002:**
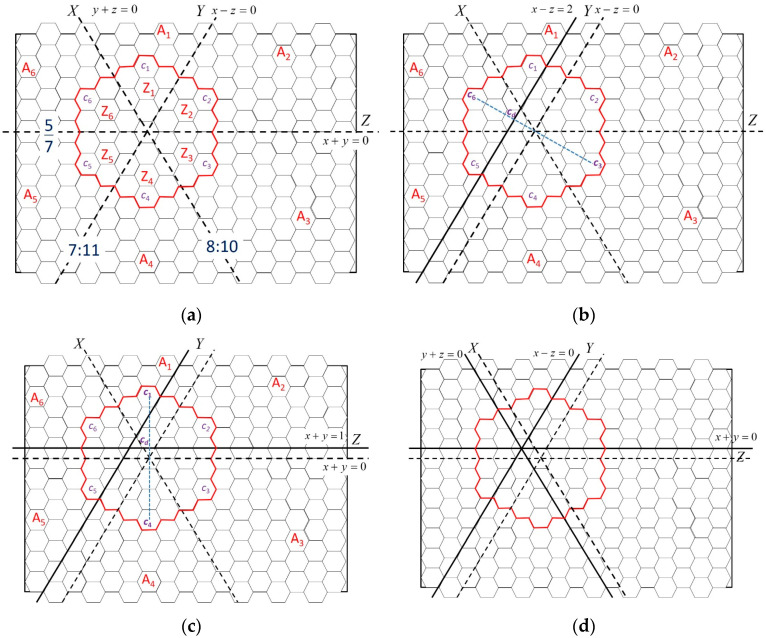
(**a**) Initial origin at the center of the charging zone; (**b**) Y-axis is adjusted to satisfy energy requirements; (**c**) Z-axis is adjusted to satisfy energy requirements; (**d**) X-axis is adjusted to orientate the origin, and the axes are fixed.

**Figure 3 sensors-23-03438-f003:**
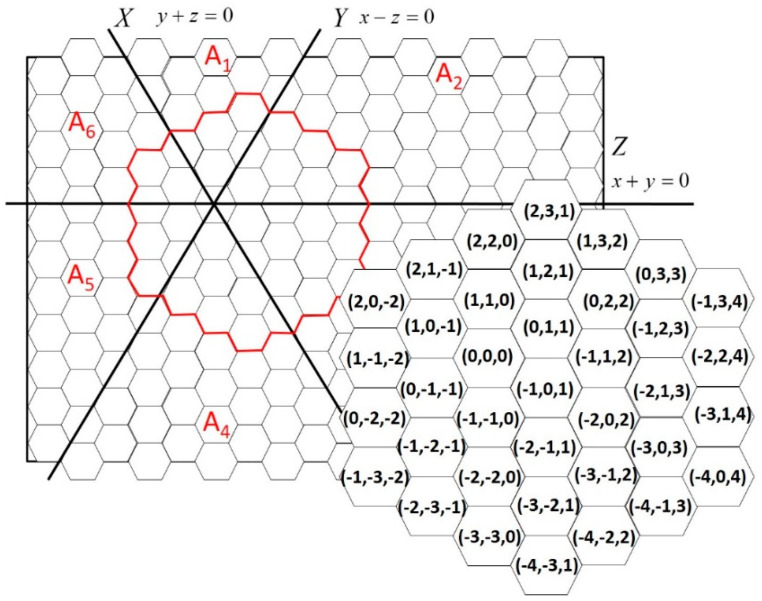
Coordinates of cells in the charging zone.

**Figure 4 sensors-23-03438-f004:**
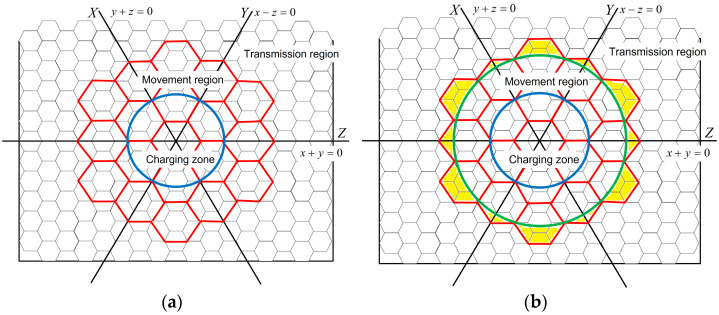
(**a**) Two coordinate systems overlapping at the origin; (**b**) Intersection of the movement and transmission regions.

**Figure 5 sensors-23-03438-f005:**
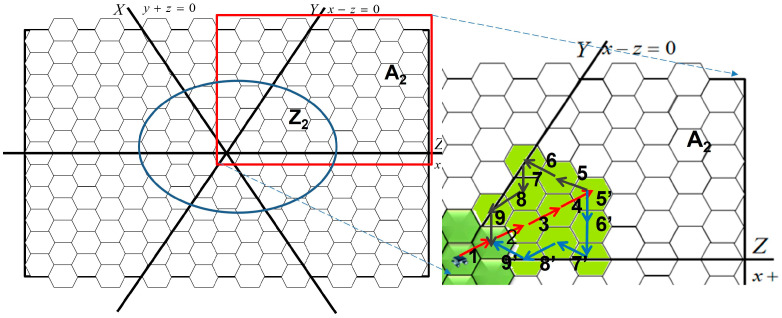
Moving paths in the movement region Z_2_. The numbers represent the steps of sensors. The sensors move in the path iteratively. For example, one sensor moves in path 1->2->3->4->5->6->7->8->9, and its following sensor should move in path 1->2->3->4->5’->6’->7’->8’->9’. The brighter green cells represent the charging zone.

**Figure 6 sensors-23-03438-f006:**
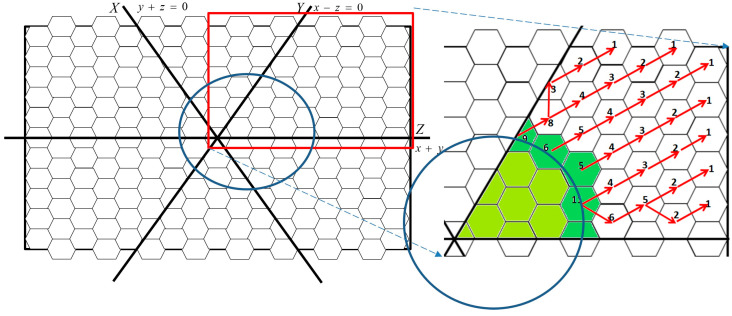
Directions and amount of energy being transmitted downstream. The numbers represent the amount of transmitted energy. The green cells are in the movement region, in addition, the intersection of movement and transmission regions is in those darker green cells.

**Figure 7 sensors-23-03438-f007:**
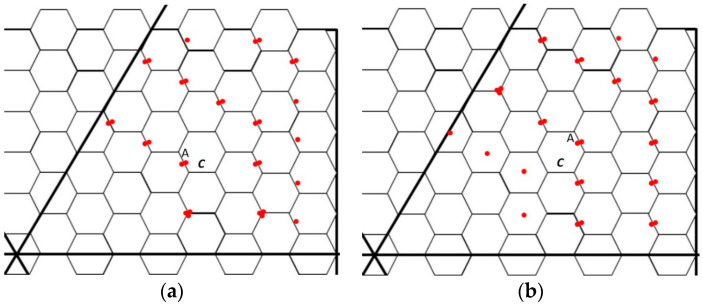
(**a**) Energy transmission in odd time slots. Sensors are represented by red dots. Sensor A in cell *C* moves to its upstream sensor and receives energy from it; (**b**) Energy transmission in even time slots. Sensors are represented by red dots. Sensor A in cell *C* moves to its downstream sensor and transmits energy to it.

**Figure 8 sensors-23-03438-f008:**
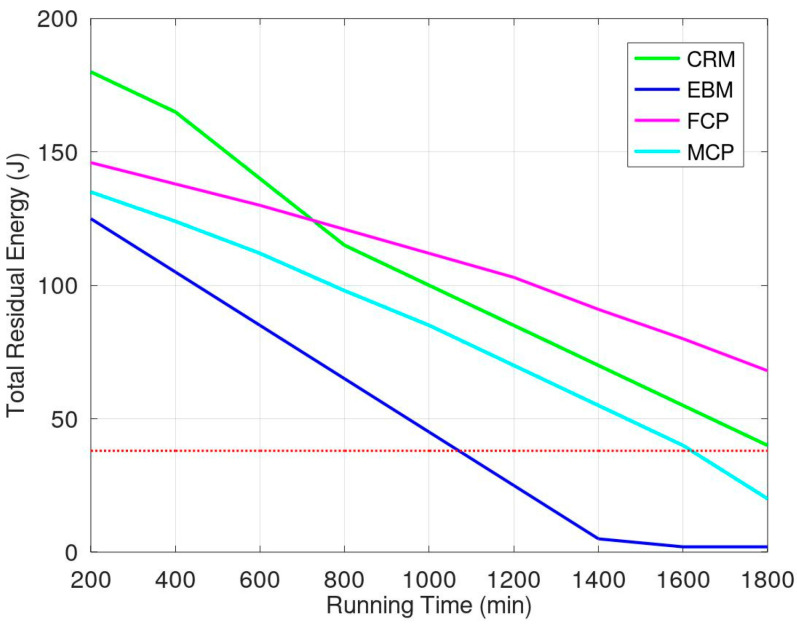
Energy evaluations related to our proposed mechanism and two previous mechanisms.

**Figure 9 sensors-23-03438-f009:**
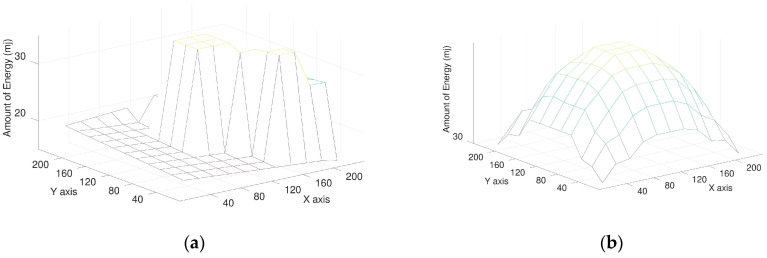
(**a**) a snapshot of energy deployment with MCP; (**b**) a snapshot of energy deployment with FCP. The least in MCP is 15 mJ, and the least in FCP is 30 mJ at the same time.

**Figure 10 sensors-23-03438-f010:**
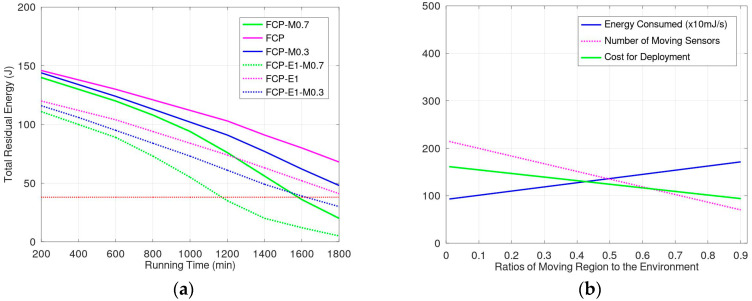
(**a**) Energy evaluations under different energy support condutions and ratios of the movement region to the environment; (**b**) Energy consumed and number of sensors according to the ratio of the movement region to the environment.

**Figure 11 sensors-23-03438-f011:**
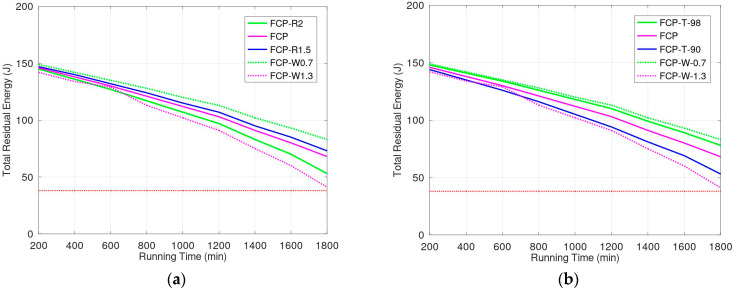
(**a**) Energy consumption of cells of different sizes and sensors of different weights; (**b**) Energy consumption from energy transmission among sensors.

**Table 1 sensors-23-03438-t001:** Pseudocode for the FCP.

Proc. FCP
Phase 1: Determine the Coordinate Systems determine the ranges of movement and transmission regions update the coordinates of movement region Phase 2: Define Rules determine movement paths in movement region determine transmission routes in transmission region Phase 3: Repeat actions of each sensor <movement> move a step following defined path if it is in charging zone stay and harvest energy else if it is in the border of regions stay and transmit energy <transmission> if it is its turn to harvest energy move to its upstream sensor stay and harvest energy if it is its turn to transmit energy move to its downstream sensor(s) stay and transmit energy

**Table 2 sensors-23-03438-t002:** Evaluation Parameters.

Parameter	Value
Size of network field	200 m × 200 m
Initial energy for total sensors	200 J
Energy consumed for monitoring events	10–20 mJ/s
Total amount of energy supply in charging zone	1–1.5 J/s
Weight of a sensor	150–250 g
Size of cells in movement region	15–30 m/edge
Size of cells in transmission region	10–20 m/edge
Energy transmission efficiency	90–98%

## Data Availability

Not applicable.

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
