# Peer review of "Charging Protocol for Partially Rechargeable Mobile Sensor Networks"

_sensors, 2023, doi:10.3390/s23073438_

Round 1
Reviewer 1 Report
The work presented in your paper describes a charging protocol oriented to partially rechargeable mobile sensor networks, which is a timely and interesting subject. Indeed, the increasing number of sensor networks deployed in different environments (in the frame of the IoT as an example) and the energy limitations to face up in this kind of networks oblige to develop innovative solutions. However, in my opinion, in its current state, your paper cannot be accepted for publication, even if the presented work and results are promising.
First, regarding Section 2 “Related Studies”, I think that a more detailed description of previous or existing works about MSNs must be provided in order to better understand the novelty of your solution. In lines 112-114, when you cite Yi and Yun [17] to give some values of transmission efficiency, the conditions allowing these values are not presented (power level, antenna efficiency, frequency of the transmitted signal…).
In Section 3, in lines 245-246, it is said that “To prolong the lifetime of a WSN, the energy in the supply zone should be transmitted to recharging zones”. Could you give more details concerning how is this energy wirelessly transmitted, what RF protocol is considered, and what is the main influence of this transmission technology to your proposed solution? Later, in lines 316-317, you propose the “near‐field radio frequency (RF) transmission technology” as a possibility, but this solution impacts the maximal distance between two nodes. How could this choice be adapted to real applications?
Concerning the Section 4, I suggest enhancing the description of the protocol depicted in the figures: to highlight the position of A2, the sequence of moving paths in A2 (presented in Figure 5). At the end of this Section, it is pointed up that “the time slots in this paper are measured in hours”. Is this time scale adequate for practical applications, not only for simulations purposes?
In Section 5, unless I am mistaken, there is no information about the platform employed to conduct the presented simulations. In Table 2, the evaluation parameters are shown but, what will be the obtained results if these parameters change? In lines 506-507, you present the values given in reference [17]. In my opinion, it should be also mentioned the kind of antennas used (gain, directivity), the frequency (wavelength) of the applied transmission signal. To measure the performance of your solution, you compare it with CRM and EBM mechanisms: could you give more details to justify this choice? Finally, in lines 585-588, it is said that “With advancements in technology related to material and production engineering, implementation of the protocol can be improved in the future”. What about the sensor platforms whose weight cannot be reduced? Could you bring some examples of possible actual existing applications?
To conclude, in Section 6, I think that the main limitations of the proposed protocol must be analyzed, comparing it with other solutions issued from the scientific literature.
Author Response
Thanks for your valuable suggestions.

Reviewer 2 Report
The paper “Charging Protocol for Partially Rechargeable Mobile Sensor Networks” provides, on the one hand, an overview of a flexible charging protocol in which energy can be transmitted from certain energy supply regions to other regions in a wireless sensor networks (WSNs) for which in order to share the energy in a certain region, the sensors can move. On the other, this paper can contribute to designing the applications that uses this type of sensors. An important application is the one proposed by the authors and developed in this paper, namely monitoring of the environment. But, as the authors themselves says, “In the future, we will implement mechanisms for matching cases using an algorithm to extend the lifetime of partially rechargeable sensor networks”. Also, a verification experiment is not carried out to prove the feasibility of the protocol. The paper remains at this level.
Following the review of the paper "Charging Protocol for Partially Rechargeable Mobile Sensor Networks ", we can conclude that:
1. A correctly and also complete formulated nomenclature, which contains all the physical quantities involved in the paper and also all the abbreviations along with the measurement units in the international system is necessary to be introduced in the paper. Thus, the readers can more easily clarify physical quantities involved in the paper;
2. The conclusions of the paper must be comprehensive and well-organized information; the paper contains many valuable results that need to be highlighted in the conclusions. In my opinion, it is necessary to add in the conclusions paragraph more relevant aspects that is presented in the present version of the paper. The conclusions of the paper must contain the possible implications of these study in future practical developments. What are the prospects for capitalizing on this research? It is necessary to add and these aspects to the conclusions.
3. In general, it is elegant that a paragraph does not end with a figure, but with a comment. In this regard, is necessary to add a comment after Figure 4 and Figure 10;
4. Please update the order of the reference in the whole text. Please make sure the first citation of all references is mentioned in numerical order;
5. In the first part of the paper, we are told that a suitable arrangement of antennas can be used for the transmission of data or energy. Then, the authors specify in paragraph 3.1. that: “Each energy supply can charge sensors within charging zones, and these sensors can transmit this energy to other sensors to extend the lifetime of the sensor network. We do not specify the energy source, but it can be, for example, solar or wind power”. The authors should to unify within the paper these concepts, in a more comprehensive concept;
6. The entire paper is based on a loading protocol, paragraph 4, which the authors assumes that: „The sensors in charging zones harvest energy from energy suppliers and that sensors in recharging zones harvest energy from neighbors. Furthermore, it was assumed that sensors harvest energy more than they consume, either from movement or energy transmission when they are in charging zones. Moreover, when a sensor transmits energy to its neighbors, it must retain energy for its continued operation, and propagation loss results in the amount of energy obtained is less than the transmitted energy. In this study, we considered building a general model which transmits energy from the charging zone to the recharging zone”. Please reformulate the statement of the charging protocol, so that it explicitly highlights the loss of energy due to the performance of the sensors' basic function, namely data transmission.
Author Response
Thanks for your valuable suggestions.

Round 2
Reviewer 1 Report
Thank you for your corrections and explanations following my comments. The current paper has been substantially improved compared with your previous version, and most of the main issues have been correctly addressed.